# DeepDISE: DNA Binding Site Prediction Using a Deep Learning Method

**DOI:** 10.3390/ijms22115510

**Published:** 2021-05-24

**Authors:** Samuel Godfrey Hendrix, Kuan Y. Chang, Zeezoo Ryu, Zhong-Ru Xie

**Affiliations:** 1Computational Drug Discovery Laboratory, School of Electrical and Computer Engineering, College of Engineering, University of Georgia, Athens, GA 30602, USA; samuel.hendrix25@uga.edu (S.G.H.); Zeezoo.Ryu@uga.edu (Z.R.); 2Department of Computer Science and Engineering, National Taiwan Ocean University, Keelung 202, Taiwan; kchang@ntou.edu.tw; 3Department of Computer Science, Franklin College of Arts and Sciences, University of Georgia, Athens, GA 30602, USA

**Keywords:** deep learning, protein–DNA interaction, binding site prediction, drug design, convolutional neural network, proteome, systems biology

## Abstract

It is essential for future research to develop a new, reliable prediction method of DNA binding sites because DNA binding sites on DNA-binding proteins provide critical clues about protein function and drug discovery. However, the current prediction methods of DNA binding sites have relatively poor accuracy. Using 3D coordinates and the atom-type of surface protein atom as the input, we trained and tested a deep learning model to predict how likely a voxel on the protein surface is to be a DNA-binding site. Based on three different evaluation datasets, the results show that our model not only outperforms several previous methods on two commonly used datasets, but also demonstrates its robust performance to be consistent among the three datasets. The visualized prediction outcomes show that the binding sites are also mostly located in correct regions. We successfully built a deep learning model to predict the DNA binding sites on target proteins. It demonstrates that 3D protein structures plus atom-type information on protein surfaces can be used to predict the potential binding sites on a protein. This approach should be further extended to develop the binding sites of other important biological molecules.

## 1. Introduction

DNA carries genetic information about all life processes, and proteins perform many essential functions for maintaining life. Interactions between proteins and nucleic acids play central roles in a majority of cellular processes, such as DNA replication and repair, transcription, regulation of gene expression, degradation of nucleotides, development (growth and differentiation), DNA stabilization, and immunity/host defense [1,2]. Moreover, the processes controlling gene expression through protein–nucleic acid interactions are critical as they increase the versatility and adaptability of an organism by allowing the cell to produce proteins when they are needed. However, revealing the mechanisms of protein–nucleic acid binding and recognition remains one of the biggest challenges in the life sciences [1,2,3,4]. Identifying the potential binding sites and residues on proteins is essential to understanding the interactions between proteins and their binding nucleic acids. A reliable prediction method will address this critical need and influence subsequent studies.

Protein binding site prediction is a critical research infrastructure, which has direct applications in drug discovery and targeting. Although numerous complex structures comprising of proteins and their binding partners, including protein–nucleic acid complexes, have been described in the public domain, many existing nucleic acid binding site prediction methods only utilize sequence (evolutional) data or residue propensities and have not yet achieved sufficient accuracy [5,6,7]. Statistical analysis of nucleic acid binding residues has helped researchers to understand the binding propensities of 20 amino acids [8,9,10,11]. However, molecular binding and recognition is a sophisticated process and is affected not only by the composition of amino acids. The subtle changes of main chain and side chain atoms, and their relative positions, change the local chemical environments on the protein surfaces. Previous studies which performed large-scale assessments of nucleic acids binding site prediction programs [5,6] also demonstrated that structure-based predictors often show better performance than their sequence-based counterparts. However, neither approach has yet achieved a satisfactory level of prediction. The sensitivity of most of the prediction methods range from 0.2 to 0.6, as some methods may have lowered their specificity to increase their sensitivity; therefore, their highest Matthews correlation coefficient (MCC) value is about 0.3 [5,6,7]. On the contrary, the methods used to predict small molecule binding sites have demonstrated sensitivity and specificity over 0.8, and their highest MCC is around 0.8 [12,13].

The reason that the accuracy of nucleic acid binding site prediction is relatively low compared to small molecule binding site prediction can be explained as follows: (1) Small molecules tend to bind to the largest cavities on the protein surface, based on the observations of previous studies [12,14]. Therefore, prediction methods which have employed the geometrical features of proteins or combined them with other chemical or energy features have often produced reliable results [14,15,16,17,18,19,20,21,22,23]. On the other hand, DNA is a long-stretched molecule and binds to relatively flat surfaces on proteins. It is less useful to apply geometrical data in nucleic acid binding site prediction than in small molecule binding site prediction. (2) The definition of a nucleic acid binding residue has not yet been standardized, and there are several definitions [5]. Different cutoffs ranging from 3.5 Å to 6.0 Å have been used to define “binding residues” [4,9,24,25,26,27,28,29,30,31,32,33,34,35,36,37,38,39,40,41]. A previous study demonstrated that a distance cutoff of 6.0 Å leads to a two-times-higher number of binding residues than that obtained with a cutoff of 3.5 Å [5]. The inconsistent cutoffs make it very difficult to evaluate, compare and improve the performance of different methods. (3) The energy-based approach has not been employed for nucleic acid binding site prediction and the binding affinities between proteins and nucleic acids has not been considered. (4) Often, DNA and RNA binding site prediction methods are developed separately [5,6]. Although DNA and RNA binding proteins usually perform different functions *in vivo*, DNA and RNA are two highly similar molecules. Their binding surfaces and binding mechanisms may be highly similar to each other [6,42]. In other words, considering RNA-binding residues/surfaces as non-DNA binding residues/surfaces or considering DNA-binding residues/surfaces as non-RNA binding residues/surfaces may interfere with the training and prediction processes.

In recent years, deep learning has been attracting attention. These methods, which generally differ from past statistical methods, do not rely heavily on human-designed hyperparameters such as feature weighting, combinations, etc. Instead, such relationships and architectures emerge after periods of training. Neural networks have shown great promise in other domains, such as the object detection and classification performed by AlexNet in the 2012 ImageNet competition [43]. Other researchers have applied similar network topologies to the problem of binding site prediction in the past with good success [44]. A common limitation to many of these approaches is that they rely on multi-layer perceptrons (MLPs) at some stage in their network. MLPs are the conventional neural network type and are essentially groups of neurons (represented through matrix operations) that connect to each other and “fire” in relation to a linear combination of their connections, often paired with a final non-linear function. The major drawback of the conventional neural network is that the input data size must be exactly the same for all data, both in training and in inference [45]. This is because they are represented by multiplying an input in the form of a matrix (the number of samples by the number of input features) by a weight matrix (the number of input features by the number of output features). Finally, in most cases a bias matrix (the number of samples by the number of output features) is then added to the output. Both operations are therefore flexible with respect to the number of samples used (weights and biases can simply be copied to form the correct matrix size), but the number of features must remain constant. Therefore, images must be resized prior to input into networks such as AlexNet. Although this is not a major issue for 2D images, which typically can be resized without significantly changing the information represented, a general method for resizing 3D graphs such as protein complexes without the risk of changing the information does not exist. This means that models using MLPs must instead only crop the data, thus creating barriers to information flow across the cropped regions. The goal of this study, like others before it, is to develop an efficient method by which a large portion of the initial pool of candidates can be screened out prior to the more expensive steps in the aforementioned pipeline.

## 2. Results

### 2.1. Model Statistics and Prediction Outcomes

The purpose of this study is to develop a deep learning model for DNA binding site prediction. After the training was completed, the prediction outcomes were retrieved and the performance of our prediction model was calculated on the training dataset (Table 1) and two external test sets (i.e., PDNA62 and PDNA224, Table 2 and Table 3). The two test sets are not totally independent of the training sets. Based on the sequence alignment outcomes (see Appendix A), there are 22 and 97 entries in PDNA62 and PDNA224, respectively, which may be homologs (sequence identity > 40%) of one or more entries in the training sets (see Appendix A). However, the performance of our prediction model on the entire test sets and the non-redundant sets (i.e., excluding the homologs) shows no significant differences (see Table 2 and Table 3). This demonstrates that our prediction model is robust.

In order to avoid overfitting during the training process, the training data were split into a 9:1 training and validation set (see Materials and Methods). The model performs well overall, with an MCC of 0.584 (Table 1). As expected, it performs slightly better on the training subset than the validation subset, but the overall performance on the validation set, with an MCC of 0.558, is still satisfactory. One observation is that the model generally performs best on “medium”-sized complexes (Figure 1). This may be because the relatively limited and rugged binding surfaces of small DNA-binding proteins and complexes are difficult to recognize through deep learning. Moreover, most complexes are in the “medium size” category, which means that the deep learning model “learns” the patterns of medium sized proteins the best. It also should be noted that the very large complexes could not be assigned to the training set but were placed in the validation set due to memory constraints.

Figure 2 shows the prediction results for the intron-encoded homing endonuclease I-Ppol (PDB ID 1a73). The prediction outcomes of the deep learning model are continuous numbers between 0 and 1, colored from blue to red. In the left panel of Figure 2, the heat map basically located most of the DNA binding surface grids, with a few false positives and false negatives. To produce binary outcomes and reduce false positives and false negatives, DeepDISE performs a clustering step. The results for 1a73 are shown in Figure 2. Although it did not achieve 100% accuracy, the algorithm largely predicted the binding area and provided hints for further research and drug design.

### 2.2. Comparing Performance

In comparison with other methods, DeepDISE was tested against the PDNA62 and PDNA224 datasets, which have been used by previous studies. As shown in Table 2 and Table 3, DeepDISE outperforms other existing methods in terms of accuracy, specificity, precision and MCC values, except its sensitivity is lower than that of two other methods. However, according to the visualization results, the “false negative” grids were not totally undetected but rather were predicted with relatively lower scores. It also needs to be noted that, unlike previous studies, our model was trained using another dataset independently of these two datasets, PDNA63 and PDNA224. Comparing our prediction results, as shown in Table 1, Table 2 and Table 3, the prediction performance was very consistent, rather than showing dramatic decreases from one dataset to another, which demonstrates that our model does not present the issue of overfitting.

## 3. Discussion

The key innovation of this study is the use of a network topology that does not require the standardization of data input. This is accomplished by using a fully convolutional neural network architecture. Convolutional network layers were originally designed to address one disadvantage of MLPs when applied to images—MLPs do not share “insights” with other neurons in the same layer. This means that when applied to images, there will almost certainly be redundant relationships stored in the network and if patterns do not appear in the exact location as in the training set, the network will not be able to recognize them easily. Convolutional layers solve this by using a set of “filters” that are convolved over the input data, creating (in most cases) an output that is the same dimension/size of the input except for the feature dimension. Thus, such layers are used in networks like AlexNet. Moreover, convolutional layers are perfectly capable of solving “segmentation” problems in which the desired result is a region of points. Given that binding site predictions can be easily formulated in this way, we proposed that a fully-convolutional network would likely achieve more desirable results than prior projects.

Deep learning algorithms have been successfully applied to image recognition. Although a few previous methods used both sequence and “structure” features, including DSSP (secondary structure), accessible surface area (ASA) and the number of H-bonds and B-factors, these features are mostly one-dimensional (i.e., features highly related to amino acid sequence). However, the input data of our model is four-dimensional (3+1, the 3D coordinates plus the atom type). This exploits the strength of a deep learning algorithm in 3D image processing and leads to the outperformance and robustness shown by DeepDISE in different datasets. Atom type alone may contain many integrated physicochemical properties, such as polarity, charge, and hydrophobicity; however, adding secondary structural information and sequence conservation to the input data may further boost the accuracy of the prediction.

Figure 3 shows the prediction outcomes for 2xma. In this case, DeepDISE achieved a prediction accuracy of 0.839, sensitivity of 0.624, specificity of 0.963, precision of 0.905 and an MCC of 0.651. The DNA-binding surfaces were mostly correctly identified, with a wide score range, illustrated by blue and red colors. Although based on the grid count, the prediction accuracy is far from perfect, the purpose of identifying the DNA-binding site was achieved. We still need to develop a better clustering algorithm to precisely group adjacent medium- to high-scored grids together in the proposed binding surface. Moreover, some false-positive grids on the protein surface may be able to bind or attract DNA molecules distantly, but the potentially-bound DNA is not shown in the PDB structure because the interactions are not strong enough to stabilize the binding of the DNA 3′ or 5′- terminals in the crystalized protein–DNA complex. This issue should be further investigated in the future.

## 4. Materials and Methods

The data pipeline began with a publicly available list of PDB files, containing both proteins and DNA. Using this source eliminates the need to hand-curate thousands of PDB files by removing duplicates, low accuracy positions, etc. The PDB files were parsed into custom format files and those erroneous ones were removed (Figure 4). These intermediate files were then processed by a C++ program that used the FreeSASA library to determine which atoms were located on the surface of the protein and to classify them according to atom types. The resulting outcomes were then fed to a preprocessor Python program that converted the atoms into a 4D Numpy array (3 spatial dimensions in 1-Å voxels plus a 1-hot encoded vector representing atom type including non-surface). For training purposes, a 3D “ground truth” array was also generated to indicate whether the locations were the binding region or not. These Numpy arrays were ultimately passed into the main Python program for training or inference using the DeepDISE model, resulting in a final prediction Numpy array. This prediction array represented a continuous heat map of where the model predicted the binding region was. For the purpose of calculating the final accuracy, a final Python program ingested the prediction array and applied k-means clustering to classify each point as binding or non-binding.

### 4.1. PDB Entries

To train our deep learning model and to test and compare its prediction accuracy with that of other existing methods, we needed two datasets of protein and DNA complexes. We obtained a PDB list of 560 DNA-interacting proteins from a manually curated database, ccPDB 2.0 [47]. The 560 PDB files were initially collected via a Python script and we automatically fetched PDB files from the RCSB database [48]. Then the same package allowed us to parse the downloaded file into a Python dictionary object for ease of use later in the pipeline. The protein complexes were then screened to insure that they contained only atoms that we could type and which had DNA within them. Finally, 274 PDB files were then saved in a custom format that allowed for an easy interface with the rest of the programs in the pipeline. In addition, we also downloaded two datasets, PDNA62 and PDNA224, consisting of 62 and 224 complexes, as two test datasets in order to test our model and compare the performance of our algorithm with that of existing ones.

### 4.2. Atom Classification and Atom Type Assignment

The PDB files were parsed, and passed to a C++/CUDA executable for atom classification. During this step, protein atoms were represented in isolation from DNA atoms to allow for solvent accessible surface area calculations to occur using the FreeSASA library. Using this process, protein atoms were classified as either surface or non-surface. Next, all surface atoms were further classified as one of 16 different atom types based on atom and residue names. These atoms were then recombined with the DNA atoms to export.

Proteins are generally made of a few elements (i.e., carbon, nitrogen, oxygen, sulfur and hydrogen). Simply classifying protein atoms into different element groups ignores their bonding and chemical environment. A common approach designed to augment the prediction performance is to label atoms not based on element alone, but also by other features such as bond order, (partial) charge, parent residue, etc. In previous studies, we developed an atom type classification scheme to describe protein–ligand interactions with a total of 23 atom types, of which 14 were for protein atoms and 20 for atoms on other ligands, with many of them shared by both [49,50]. In order to improve the performance of this classification scheme to avoid assigning chemically dissimilar atoms into the same atom type (e.g., nitrogen located on the main chain and the histidine side chain), we made some modifications and used it as the basis to create a new nucleic acid prediction method. As shown in Table 4, the atom types are identified by a 3-code or a 3-letter name for those that do not need to be further classified, because they are relatively rarely observed in our datasets of protein-ligand complexes (i.e., metals and phosphorus). Some general rules for the 3-code names were as follows: the 1st code is the name of the element (C, N, O, and S) and the 2nd and 3rd codes indicate the surroundings and electrostatic properties of the atom. The 2nd code can be 2, 3, R, or C, which, respectively, correspond to sp2 or sp3 hybridization or inclusion in an aromatic ring or conjugated system. The 3rd code can be N, P, V, or C, which respectively correspond to a nonpolar, polar (can be a hydrogen bond donor or acceptor), variable or charged atoms. The “variable (V)” code is associated with the atom type NRV, which is used primarily for the two nitrogen atoms on the imidazole ring of a histidine, as both of the nitrogens can be either protonated (hydrogen bond donor) or deprotonated (hydrogen bond acceptor). For simplicity, the nitrogen of tryptophan, which is more infrequently seen than histidine, especially in the active sites, was also assigned the atom type NRV. We developed an algorithm which automatically assigned an atom type to each protein atom. To assign an atom type to each atom on a binding complex, we need to know the element, the bond orders that connect the atom to others, and which atoms it connects to. Based on our knowledge about the nomination system and structure of common amino acids, we had all the bonding information we mentioned above as long as we knew the atom names and residue names and compiled them in a PDB file.

### 4.3. Preprocessing

After determining the atom types of all protein atoms, final preprocessing was performed. Upon ingestion, the script constructed a 4D Numpy array, where the dimensions corresponded to spatial dimensions x, y, and z, and an additional atom type dimension was constructed. The array was designed to be 3 Å in all 3 directions and was subdivided into 1-Å voxels. After the array was allocated, the script iterated over all the protein atoms and populated the voxel which was closest to the center position of the atom with its type, which was recorded into the array. For training purposes, distance calculations were performed to determine the binding region and generate a corresponding 3D mask. In this case, the binding region was defined as the up to 6-Å region between the center of a protein atom and a DNA atom. All voxels within this region were set to 1 and all voxels not in this region were set to 0. Finally, both the input and mask arrays were rotated into 24 unique 90° 3D rotations and were saved to a Numpy compressed archive with PDB and rotation IDs in the file name.

### 4.4. Model

The DeepDISE model project is a fully convolutional neural network written with Pytorch Lightning. Any model of which the input and output are in the form of arrays can be executed using traditional CPUs or much faster GPUs.

#### 4.4.1. Architecture

The high-level architecture of DeepDISE was based on a fully convolutional neural network called UNET [51]. Under this architecture, data entered the network and passed through a series of blocks composed of convolutional layers. For the first half of the network, the output of each block was down-sampled using pooling layers, before being passed on to the next block in the sequence. The last half of the network up-sampled the output of each block by the same ratio as the previous layer’s down-sampling. The final output of the network was the same size as the input, which led itself very well to segmentation problems, in which the output needed to act as a mask on the input. It has been theorized that UNETs perform well relative to other architectures since they have multiple scales for the convolutional operations to act on, in contrast to ResNets, and multiple pathways for the gradient to flow through, in contrast to simple feedforward networks.

Within each block of the network, a separate architecture was implemented based on the DenseNet architecture [52]. Each layer in the DenseNet architecture was a single convolutional layer paired with an MISH activation function [52]. Under this architecture, the input of each layer was the concatenated sum of the initial input and all the outputs of the previous layers. This architecture worked under a similar assumption as the UNET models—they were more easily trained because the optimizer had clear paths through the gradients of the initial layers. In the DeepDISE model there are 4 such layers.

#### 4.4.2. Training

The DeepDISE model was trained using a curated set of PDBs with proteins binding to DNA. The PDB records were preprocessed and split into a 9:1 training and validation set. Although the training set is not totally non-redundant, the all-against-all pairwise sequence alignment results (see Supplementary Material) showed that less than 1% of pairs of sequences were homologs. Any complexes that were too large to be trained on using a Nvidia 2070 Max-Q GPU were also added to the validation set. In the training set, each protein-DNA complex was rotated 24 times to generate 24 structural files with different orientations. Among the 24 orientations, 2 were added to the validation set to better track the model’s performance while training, but they were removed prior to the final statistical calculations for this paper. The model was trained over the course of 48 h using the Ranger optimizer and binary cross entropy for the loss function. The Ranger optimizer was chosen as it has been shown to produce good results in other applications and was essentially the combination of AdamW and LookAhead. While training, real-time statistics were exported in the TensorBoard format via a Pytorch Lightning callback so that we could determine the model’s overall convergence and spot issues without need to wait for training to be fully completed.

In Appendix A, there is a raw representation denoted by light blue and a “smoothed” representation denoted by dark blue in each image. The training error essentially converged, but the validation error slowly decreased over time. This was expected, as the training data points represent individual proteins, whereas the validation data points represent the full validation set, thus averaging out the perceived variance.

Training was stopped after roughly 2.5 epochs of going through the training data (validation was calculated every 1/8th epoch). The final binomial cross-entropy score across both the training and the validation sets together was 0.02358, compared to 0.69315 if the model had only predicted non-binding for the full dataset (the most common true prediction for points).

### 4.5. Clustering and Final Prediction

The DeepDISE model creates a continuous output. For applications where a binary classification is needed, an additional step to generate the final prediction is required. Initial experiments with linear classification were explored, but ultimately the accuracy did not seem to align with the qualitative results of the model output. Because of this, we decided to use a system based on k-means clustering. This allowed for the binding site determination not only to leverage the prediction score given by the model, but also the spatial location relative to other scores as well. To arrive at a binary classification, two rounds of clustering were used. In the first round, n clusters were generated, where n equals the number of atoms divided by 1000, rounded up to the nearest integer. The clustering algorithm was then shown the list of points comprising the prediction, where all the values were first standardized then the score dimension was increased by 5× to bias the clustering towards it. The n clusters were then passed to a second round of clustering, where the algorithm was only given the average score of each cluster and was required to cluster them into 2 clusters. Finally, the cluster of clusters with the highest average score was labeled the binding cluster and all points within it were assigned to the binding region.

### 4.6. Assessment of the Binding Site Prediction

The final statistics were computed on a per-grid basis, where each grid represented a 1-Å voxel within the complex. Ground truth was determined by labeling each voxel as either binding or non-binding as a function of its proximity to both a DNA atom and a protein atom. For each voxel, the algorithm first iterates over each protein atom in the complex. If the atom is within 6 Å of the voxel, the algorithm then checks if the voxel is within 6 Å from a DNA atom. If this is the case, it finally checks to see if that DNA atom is also within 6 Å of the protein atom from the first step. If this is the case, the voxel is labeled as part of the binding site.

## 5. Conclusions

In this study, we have developed a deep learning-based method to model and predict the DNA binding sites on target proteins. Due to its robustness, this model can be applied to different datasets to identify the potential DNA-binding sites of most of the target proteins successfully. We have also demonstrated that by using only the 3-dimensional protein structures plus the assigned atom type on the surface atoms, we were able to train a deep learning model to predict DNA binding sites. This approach should be also applied to create models to predict other binding partners of a target protein, such as medical compounds or other proteins. When we build up all these prediction models and integrate them together, we will be able to detect all the functional patches on a target protein and further reveal the recognition mechanism of our proteome.

## Figures and Tables

**Figure 1 ijms-22-05510-f001:**
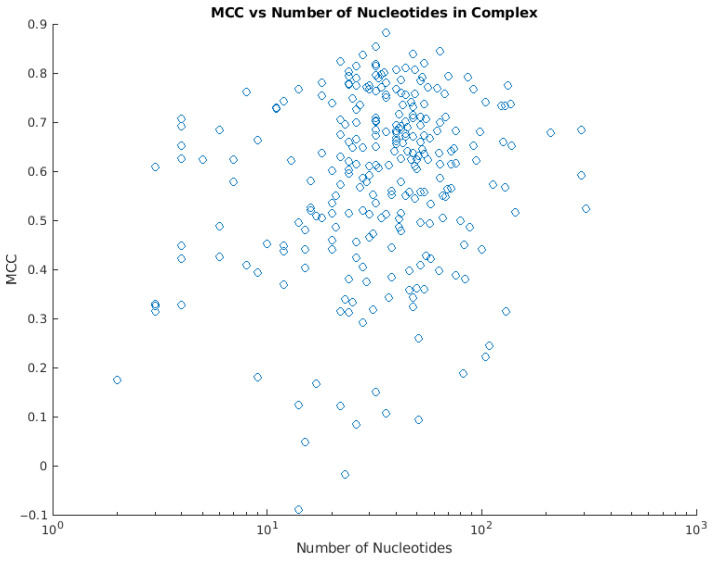
The relationship between the number of binding nucleotides and the MCC value of each complex.

**Figure 2 ijms-22-05510-f002:**
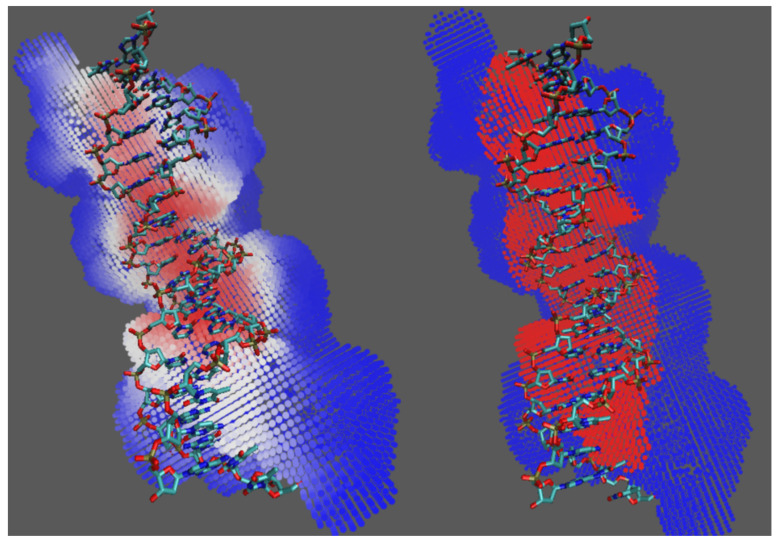
The prediction outcomes for protein PDB ID 1a73 before (**left**) and after (**right**) clustering.

**Figure 3 ijms-22-05510-f003:**
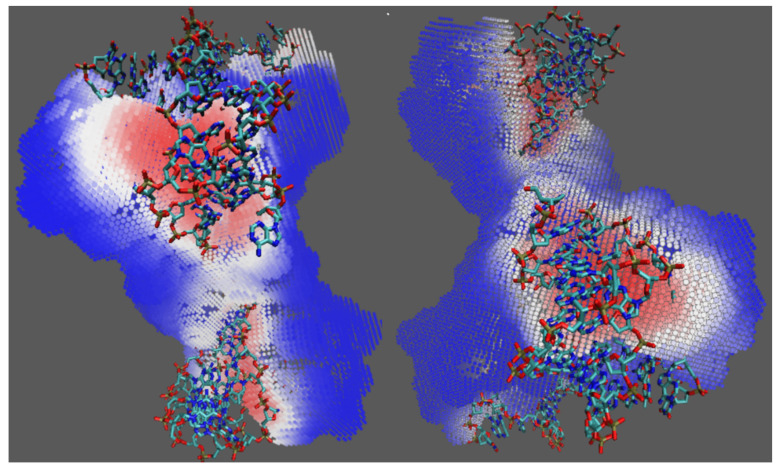
Prediction results for 2xma. The structure was rotated around 150 degrees and presented parallelly.

**Figure 4 ijms-22-05510-f004:**
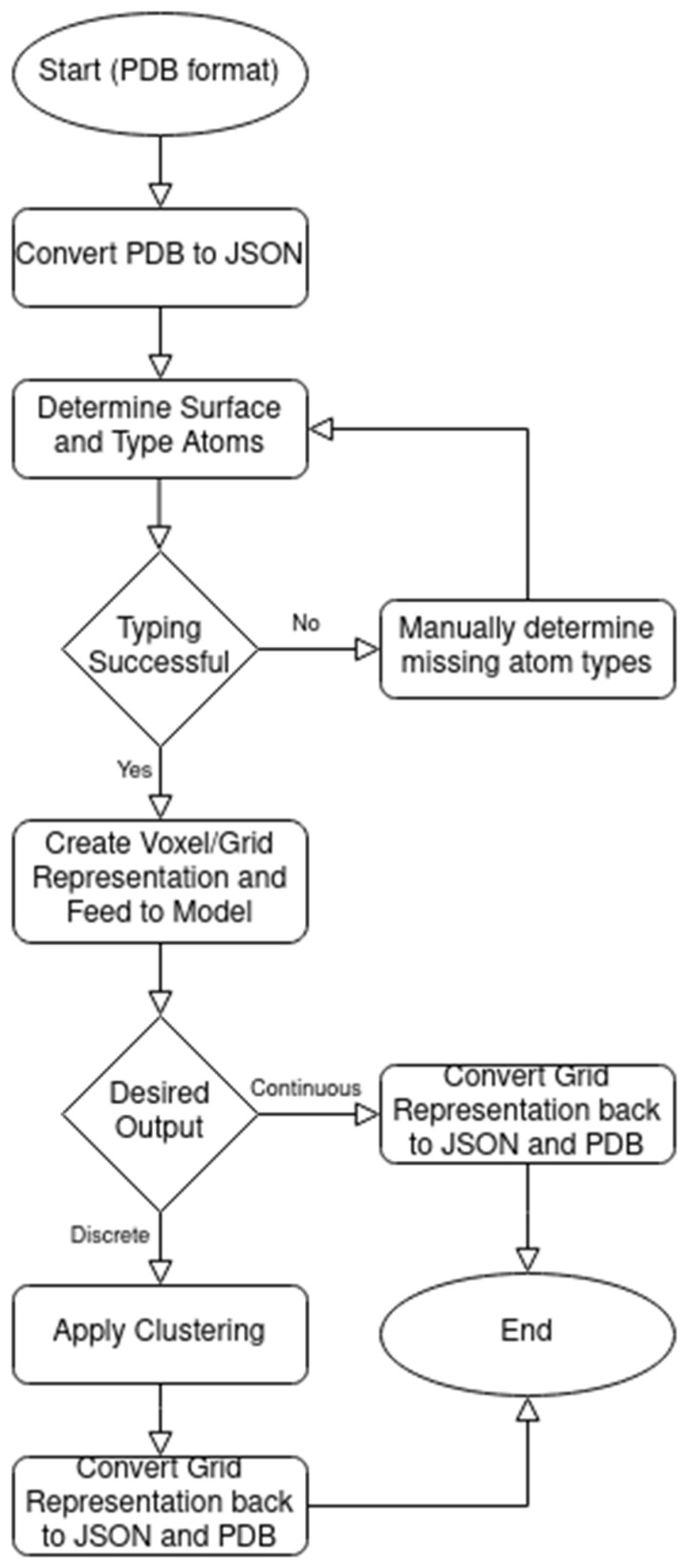
Flowchart of DeepDISE.

**Table 1 ijms-22-05510-t001:** Prediction performance of DeepDISE on the training and validation sets.

Methods	Accuracy	Sensitivity	Specificity	Precision	MCC	AUC
Training	0.877	0.738	0.907	0.606	0.586	0.926
Validation	0.884	0.691	0.924	0.600	0.558	0.928
Full Dataset	0.878	0.734	0.908	0.606	0.584	0.927

**Table 2 ijms-22-05510-t002:** Performance of DeepDISE compared with previous methods using PDNA62 [46].

Methods	Accuracy	Sensitivity	Specificity	Precision	MCC	AUC
Dps-pred	0.791	0.403	0.818	0.279	0.191	-
Dbs-pssm	0.664	0.682	0.660	0.210	0.249	-
BindN	0.703	0.694	0.705	0.291	0.297	0.752
Dp-bind	0.781	0.792	0.772	0.378	0.490	-
BindN-RF	0.782	0.781	0.782	0.385	0.436	0.861
BindN+	0.790	0.773	0.793	0.395	0.443	0.859
PreDNA	0.794	0.768	0.797	0.398	0.424	-
EL_PSSM-RT	0.808	0.854	0.801	0.428	0.507	0.901
PDRLGB	0.815	0.863	0.806	0.438	0.523	0.912
**DeepDISE ***	**0.752(0.750)**	**0.765(0.767)**	**0.905(0.908)**	**0.725(0.735)**	**0.658(0.662)**	**0.931(0.926)**

* The numbers inside the parentheses were calculated after removing redundant (sequence identity > 40%) structures.

**Table 3 ijms-22-05510-t003:** Performance of DeepDISE compared with previous methods using PDNA224 [46].

Methods	Accuracy	Sensitivity	Specificity	Precision	MCC	AUC
PreDNA	0.791	0.695	0.798	0.195	0.289	-
EL_PSSM-RT	0.781	0.796	0.780	0.203	0.341	0.865
PDRLGB	0.800	0.833	0.797	0.224	0.383	0.896
**DeepDISE ***	**0.880(0.881)**	**0.770(0.779)**	**0.901(0.898)**	**0.596(0.558)**	**0.607(0.593)**	**0.930(0.929)**

* The numbers inside the parentheses were calculated after removing redundant (sequence identity > 40%) structures.

**Table 4 ijms-22-05510-t004:** Atom types and descriptions.

Atom Type	Description
C2P	C, SP2, Polar
C3N	C, SP3, Nonpolar
C3P	C, SP3, Polar
CRP	C, Aromatic, Polar
CRN	C, Aromatic, Nonpolar
CCP	C, Conjugated/resonating, Polar
N3C	N, SP3, Charged
NRV	N, Aromatic, Variable
NCC	N, Conjugated/Resonating, Charged
NCP	N, Conjugated/Resonating, Polar
O3P	O, SP3, Polar
OCC	O, Conjugated/Resonating, Charged
OCP	O, Conjugated/Resonating, Polar
S3N	S, SP3, Nonpolar
PHO	Phosphorus
MET	Metal

## Data Availability

The data presented in this study are openly available on https://gitlab.godfreyhendrix.com/cddl/deepdise-prediction-results (accessed on 30 April 2021) and are free for academic users.

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
