# Peer review of "DeepDISE: DNA Binding Site Prediction Using a Deep Learning Method"

_ijms, 2021, doi:10.3390/ijms22115510_

Round 1

Reviewer 1 Report

The manuscript of Samuel Godfrey Hendrixet al.describes a deep learning model to predict DNA binding sites on target protein. Only 3D protein structures plus assigned atom types on the surface atoms can be used to train a deep learning model to predict DNA binding site. Accurate prediction of binding sites of ligands with a weak surface complementarity of ligand/protein interactions and with a ligand binding mediated by a large binding surface still remains an obstacle in the field of ligand protein interactions. The authors show that their deep learning model is efficient for DNA ligand and suggest extending it to develop the binding sites of other important biological molecules such as carbohydrates. But in the glycans context it is difficult to apply because few glycan protein complexes are available, especially for long-chain glycan, so I suggest removing carbohydrates in row 367.

Author Response

We thank the reviewer for the constructive comments. As suggested, we removed carbohydrates in row 367 (row 378).

Reviewer 2 Report

Authors presented a deep learning method to predict DNA binding site in PDB structures.

Major concerns:
Measures to separate the training data from that used in the evaluations are needed, the results presented are likely to be influenced heavily by over-fitting:
    1. Line 310: the division of 9:1 training and validation needs to excludes homologous, so that no structures in the validation are homologous to those in the training. 
    2. Line 113: “performance of our prediction model was calculated on the training dataset and 2 external test sets (i.e. PDNA62 and PDNA224)”. Datasets that are collected independently are not necessarily independent. The authors needs to measure the overlap between these two test datasets and the training data.

The paper is not readable, some examples:
    1. Line 96: “The major drawback for this type of neural network is that the input data size must be exactly the same for all data both in training and in inference”, please explain more and provide citation 
    2. Please reconsider lines 85 to 108.
    3. Line 223: what are “the native capabilities of the BioPython package”? Perhaps technical issues need to be in the supplement. 
    4. This also include the format of the data; converting to a JSON format (which is repeated again and again in the script) is not of concern to readers. The word JSON should not appear in the main text.

In addition: MCC is used for classifiers with a binary output. Unlike MCC, AUC measures the classifiers separation power over all cut-off values. Thus, if a classifier generates a probability value, such as CNN, it is best and more robust to contrast its performance using AUC values. 
The authors cited two review paper regarding the current predictors (references 5 and 6); both relies on AUC in contrasting performance, and only 6 also include MCC. Thus, the output of DeepDISE needs to be evaluated mainly using AUC measures.

Author Response

Our responses to the comments from the reviewer are detailed below, and all the texts added/changes made are highlighted with red font in the revised manuscript, except for the figure legends.

Authors presented a deep learning method to predict DNA binding site in PDB structures.
Major concerns:

Measures to separate the training data from that used in the evaluations are needed, the results presented are likely to be influenced heavily by over-fitting:

  1. Line 310: the division of 9:1 training and validation needs to excludes homologous, so that no structures in the validation are homologous to those in the training.

Response:

We thank the reviewer pointed out his/her concern. We performed an all-against-all pairwise sequence alignment inside the training set and found out that only few pairwise alignments had larger than 40% sequence identity (the alignment results are shown in supplementary material). As model training is time-consuming, we did not exclude the homologs in training data. However, there is no evidence showing the model is overfitting. In addition, we did compare the prediction performance on two data sets with and without homologs involved. The performance is highly similar and shows no sign of overfitting. We added explanations in line 321-323.

  1. Line 113: “performance of our prediction model was calculated on the training dataset and 2 external test sets (i.e. PDNA62 and PDNA224)”. Datasets that are collected independently are not necessarily independent. The authors needs to measure the overlap between these two test datasets and the
    training data.

Response:

Thanks again for pointing out this concern. We did perform sequence alignment between the training set and two test sets (results are in supplementary material). There are 97 (out of 224) and 22 (out of 62) entries that are redundant. However, the performance before and after we exclude the redundant PDB files are highly similar and presented in Table 2 and 3. We added some explanation in line 120-126.

The paper is not readable, some examples:

  1. Line 96: “The major drawback for this type of neural network is that the input data size must be exactly the same for all data both in training and in inference”, please explain more and provide citation

Response:

Thank you for reminding us to explain this point clearly. The input data of a conventional neural network needs to have exactly the same dimensions but a deep learning convolutional does not. We added some explanation in line 100-106.

  1. Please reconsider lines 85 to 108.

Response:

We thank the reviewer for this concern. We did not really understand the specific point but tried to revise the paragraph of line 87-114.

  1. Line 223: what are “the native capabilities of the BioPython package”? Perhaps technical issues need to be in the supplement.

Response:

We thank the reviewer for the constructive comments. As suggested, we removed some technical details from line 233-234.

  1. This also include the format of the data; converting to a JSON format (which is repeated again and again in the script) is not of concern to readers. The word JSON should not appear in the main text.

Response:

We thank the reviewer’s reminder. As suggested, we removed the word JSON from the main text and used some other terms such as an intermediate file instead. Please see line 214 and line 233-237

In addition: MCC is used for classifiers with a binary output. Unlike MCC, AUC measures the classifiers separation power over all cutoff values. Thus, if a classifier generates a probability value, such as CNN, it is best and more robust to contrast its performance using AUC values. The authors cited two review paper regarding the current predictors (references 5 and 6); both relies on AUC in contrasting performance, and only 6 also include MCC. Thus, the output of DeepDISE needs to be evaluated mainly using AUC measures.

Response:

We thank the reviewer for the constructive comments. As suggested, we added AUC of our model in Table 2 and 3.